# Reliability Evaluation of Cabled Active Distribution Network Considering Multiple Devices—A Generalized MILP Model

Jiaxin Zhang [1,2,*], Bo Wang [1,2], Hengrui Ma [1,2], Yifan He [3], Yiwei Wang [1,2] and Zichun Xue [1,2]

1   Hubei Engineering and Technology Research Center for AC/DC Intelligent Distribution Network, School of Electrical Engineering and Automation, Wuhan University, Wuhan 430072, China; whwdwb@whu.edu.cn (B.W.); henry3764@foxmail.com (H.M.); 2019302070238@whu.edu.com (Z.X.)
2   School of Electrical Engineering and Automation, Wuhan University, Wuhan 430072, China
3   Electric Science Research Institute, State Grid Zhejiang Province Co., Ltd., Hangzhou 310013, China; yifan_he@126.com
*   Correspondence: 2020282070157@whu.edu.cn

**Abstract:** With the rapid development of power electronic equipment, the automation and intelligence level of active distribution networks (ADNs) continues to improve. Against this background, soft open points (SOPs) are gradually replacing traditional segmented switches and interconnection switches. The voltage support capability and fast response characteristics of SOPs can shorten power outage time and expand load recovery range. However, the widespread integration of distributed renewable energy and new power electronic devices has made the fault characteristics of ADNs more complex, significantly increasing the computational complexity of ADN reliability assessment. At present, there are few studies that comprehensively consider ADNs with multiple devices. Therefore, this paper proposes a reliability evaluation method for ADNs that considers multiple devices. Firstly, the impact of circuit breakers, SOPs, and segmented switches on the load recovery process is analyzed. Secondly, an improved virtual fault flow model based on the action mechanisms of circuit breakers, SOPs, and segmented switches is established. The virtual fault flow is represented by logical variables to simulate post fault network reconstruction strategies that include circuit breaker tripping, SOP power supply recovery, and segmented switch isolation actions. Then, with the goal of minimizing the system average outage time after network reconstruction, a generalized mixed integer linear programming (MILP) model is given. Finally, taking the IEEE 33-node testing system as a case, the effectiveness and feasibility of the proposed method are demonstrated.

**Keywords:** active distribution network; power electronic devices; reliability evaluation method; mixed integer linear programming (MILP) model

## 1. Introduction

With the rapid development of active management technology, the form of distribution networks is undergoing a historic transformation, that is, conventional distribution networks are gradually transforming into active distribution networks (ADNs) containing a high proportion of renewable energy [1,2]. ADNs have become a hub for achieving comprehensive energy interconnection [3], playing an important role in ensuring the safe, economical, and reliable use of electricity [4]. Among them, adjustment technologies such as intelligent SOPs [5], fault indicator [6], uninterruptible power supply (UPS) [7], and user-side demand response (DR) [8] have also been widely applied. ADNs can quickly adjust the system flow through automation equipment, achieve fault location and isolation, and improve system reliability. Due to the different roles played by different devices in the process of fault recovery, when a fault occurs, the circuit breaker, the segmented switch, the SOP, and the fault indicator that detects fault current operate at different stages with different operating principles and different impacts on system reliability [9]. Therefore,

accurate ADN reliability assessment and planning need to simultaneously consider the impact of different equipment and load recovery stages [10].

Currently, there are two main types of reliability assessment methods for distribution networks: analytical methods and simulation methods. References [11–14] used a fault diffusion method to conduct a breadth search on circuit breakers and switches in order to obtain the range of fault impact. In detail, reference [11] proposed a practical reliability evaluation algorithm for power distribution systems with general network configurations. This algorithm is an extension of the analytical simulation method for radial distribution systems; the proposed method in [12] develops a comprehensive plan that specifies the optimal plan and schedule for the installation of lines, substations, and automation equipment; reference [13] proposed a novel multistage planning model for implementing a distribution automation system, considering all main hardware and software infrastructures. The aim is to provide a methodology for developing plans to optimally determine the type, location, and time in which equipment or infrastructure must be added to the network, taking into account the network capacity expansion plan. Reference [14] proposed a novel profit-oriented model for multistage distribution network expansion planning with distribution automation. The objective function to be maximized is the net present value of the company's profit. In addition, reference [15] proposed a distribution network reliability evaluation method based on a fault correlation matrix by calculating reliability indicators; reference [16] recursively calculated reliability indicators from upstream to downstream using algebraic equations, providing a new approach for evaluating reliability through algebraic operations; reference [17] obtained the failure mode consequence analysis matrix through topology identification and matrix operation, thereby calculating the reliability index. In order to simulate different operating modes, the Monte Carlo simulation method is widely applied to complex distribution networks [18–20]. The simulation method has been widely used in flexible simulation of switch actions [21], islanding operation [22], and self-healing recovery [23]. In the above reliability evaluation methods, the analytical method struggles to consider the load recovery strategy when the network topology changes, which can lead to an underestimation of distribution network reliability indicators [24]. The simulation method is computationally complex when considering circuit breakers and segmented switches, and it is difficult to directly embed the reliability calculation into the planning problem using a posterior approach.

Therefore, the distribution network reliability planning model is essentially still a mixed integer nonlinear programming model, usually solved using heuristic algorithms such as the genetic algorithm and particle swarm optimization algorithm, which cannot guarantee the global optimal solution. Compared to traditional models, the MILP model can ensure a global optimal solution, which can be solved by a commercial solver. Based on this, numerous scholars have also conducted research on this issue and achieved good results. Reference [25] proposes a reliability evaluation method based on MILP, which describes the upstream and downstream relationships of nodes through virtual power flow, and directly embeds reliability evaluation methods into distribution network planning. Given the unique advantages of MILP, reliability evaluation methods based on MILP have gradually become a research hotspot. Reference [24] proposed a distribution network reliability evaluation model based on MILP considering network reconstruction; reference [26] proposed a reliability evaluation method based on MILP, which considers the impact of circuit breaker and segmented switch actions in detail and can obtain more practical reliability evaluation results. Reference [27] considered the impact of UPS on reliability. Reference [28] analyzed the fault localization process of the fault indicator and further improved the MILP model. The reliability evaluation method based on MILP can effectively analyze the mechanism of load recovery strategies and is easily embedded in the planning and operation model. However, reliability assessment methods based on MILP usually only consider fault isolation and fault repair separately, or fault localization modeling separately, without comprehensively considering the impact of fault isolation,

fault isolation, fault localization, and fault forwarding [25–28], resulting in underestimated reliability results.

With the continuous improvement of ADNs' automation and intelligence level, electrical equipment such as a SOP that can flexibly adjust its operating status is gradually being put into use in the power grid. In terms of reliability assessment based on MILP, there is little comprehensive consideration of the impact of circuit breakers, segmented switches, SOPs, fault indicators, and UPS on the reliability of distribution networks. Therefore, this paper designs a general MILP reliability evaluation model by comprehensively considering the mechanism of these devices, providing new ideas and technical support for ADN reliability evaluation methods. The remainder of this paper is organized as follows: in Section 2, the detailed recovery process is analyzed. Based on this, an improved virtual power flow is proposed and the whole process is divided into three different stages, including right-after stage (RA), segment switch fault isolation stage (SSI) and fault location and load transfer (LT) stage. Then, by introducing intermediate variables, an improved relationship between the decision variables of virtual power flow analysis of equipment working state and reliability index variables is built in Sections 3.1–3.3. Meanwhile, an ADN reliability evaluation method based on MILP is established in Section 4. The effectiveness and feasibility of the proposed method is verified in Section 5. Finally, Section 6 concludes the paper.

## 2. Analysis of Load Recovery in an ADN

Based on the working principles of circuit breakers, fault indicators, segmented switches, SOPs, UPS, and other equipment, this section analyzes the impact of their participation in load recovery on system reliability. Among them, the circuit breaker isolates the fault current through tripping, reducing the fault diffusion range; the segmented switch isolates faults and restores upstream power supply through reverse operation; the fault indicator determines the fault current and combines it with the line inspection process to locate the fault; the UPS reduces power outage time by continuously restoring load power supply; the SOP quickly achieves power locking by switching control modes, and its DC isolation link quickly isolates the feeders at both ends. In conjunction with segmented switches, it achieves fault isolation and power recovery. In order to provide a better understanding of this process, the load recovery method is described as follows:

Step 1: When a continuous fault occurs in an ADN, the fault is isolated by tripping the circuit breaker, forming a downstream power outage area of the circuit breaker. Meanwhile, the UPS quickly restores the load of the node where it is located;

Step 2: Isolate the power outage area with segmented switches and restore some loads in non-outage areas, forming downstream power outage areas of segmented switches and upstream isolation recovery areas of segmented switches;

Step 3: The fault indicator determines the fault feeder section by flowing through the fault current and manually inspects the fault line. After the fault line inspection time, the fault is further isolated and part of the load is restored;

Step 4: Through the SOP, load transfer is achieved and the SOP fault-side converter switches from PQ control mode to Vf control mode to provide voltage support and achieve rapid load transfer in power outage areas;

Step 5: The load that cannot be restored through switch action is restored to a normal operating state through the repair or replacement process of faulty components.

In summary, the fault recovery process includes four processes: (1) Fault occurrence process; (2) Fault isolation process; (3) Fault location and load transfer process; (4) Fault repair process.

The fault occurrence process, which is the instantaneous process from the fault occurrence to the detection of a fault current by the circuit breaker, is not considered in this paper due to its short duration. The process of fault isolation, that is, the process of fault isolation and load recovery through the action of circuit breakers and segmented switches, is divided into the right-after stage (RA) and the segment switch fault isolation stage (SSI).

The fault location and load transfer (LT) stage refers to the process of locating faults through fault indicators, further isolating faults, and combining them with SOP transfer. Among them, the fault indicator determines the fault current path through fault current, reduces the range of fault line inspection, and further isolates the fault through manual line inspection; switches the SOP fault-side converter to Vf control mode; and provides voltage support. The non-fault-side converter of the SOP adopts a DC voltage-reactive power ($U_{dc}Q$) control mode to maintain stable power transmission on both sides of the converter and maintain stable DC voltage on both sides of the SOP.

The fault repair process refers to the entire process from component failure to the restoration of the system to normal operation after component repair is completed.

In this paper, when a fault occurs, the ADN still operates as a whole body and does not form an island. Therefore, the power flow constraints still need to be satisfied in the RA stage, SSI stage, and LT stage. On the other hand, the operating status of the equipment determines the power supply status of the load point. For example, the load point will be cut off when the circuit breakers and segmented switches are disconnected. Therefore, as long as the operating status of the equipment in each stage is obtained, the system reliability results can be obtained. It can be seen that the number and location of equipment in an ADN will have a significant impact on reliability indicators, so it is necessary to model its action mechanism in detail.

## 3. Improved Virtual Power Flow Considering Multi-Stage Fault Recovery Process

The widespread access of distributed power sources and power electronic devices has increased the complexity of fault characteristics in distribution networks, making traditional reliability analysis methods difficult to apply. Therefore, this article draws on reference [25] and considers the combination mechanism of circuit breakers, segmented switches, SOPs, fault indicators, and UPS to expand the virtual power flow. The improved virtual power flow simulates the state variables of equipment during the load recovery process after a fault occurs. It starts from the fault components, propagates through nodes and lines, and is ultimately isolated by circuit breakers, segmented switches, SOPs, fault indicators, and UPS. After a fault event occurs, the power outage status of a load point can be represented by node, line, and equipment state variables. Therefore, the reliability indicators of the load point can be directly calculated based on the given network topology, reliability parameters, and other data. Reliability indicators commonly used include system average interruption frequency index (SAIFI), system average interruption duration index (SAIDI), average system availability index (ASAI), and expected energy not supplied (EENS). Reliability parameters commonly used include failure rate and repair time. This section comprehensively considers the impact of the immediate power outage stage, segmented switch isolation stage, fault location, and SOP transfer stage on the load recovery process, and proposes a virtual power flow modeling method.

For greater clarity, a definition is provided and the symbol and set indices are summarized as follows: $i$ and $j$ represent the distribution network node indices; $ij$ represents the index of distribution network lines; $xy$ represents the index of line $xy$; $\gamma$ represents a set of faulty lines; NO indicates that the system is in normal operation; $f$ represents the feeder index; $h$ represents the load level index; $\gamma_I^{SOP}$ and $\gamma_J^{SOP}$ represent the collection of the first and last ends of the line ij installed SOP; $\gamma_I^{UPS}$ represents a node set with UPS; $\gamma_I^B$ and $\gamma_I^{SS}$ represent sets with installation of a circuit breaker and sets with a segmented switch at the beginning of the line, respectively. $\gamma_J^B$ and $\gamma_J^{SS}$ represent sets with installation of circuit breakers and sets with a segmented switch at the end of the line, respectively. $\Psi^F$ represents a set of feeders; $\Psi^{SUB}$ represents a set of substation nodes; $\Psi^{LN}$ represents a set of load nodes; $\Psi^L$ represents a set of lines; H represents the set of load levels.

### 3.1. Virtual Power Flow Model in RA Stage

After a fault occurs, the *RA* stage is the process of circuit breaker tripping and fault current isolation. The *RA* virtual power flow starts from the faulty components and

propagates through the state variables of nodes and lines until the first circuit breaker flows through. The fault outage range of the *RA* stage is represented by the node and line state variables. The virtual power flow model in the *RA* stage is shown in (1)–(7).

$$f_{xy}^{xy,RA} = 0 \tag{1}$$

$$-(1 - s_{ij}^{i,NO,SS})M + f_i^{xy,RA} \le f_{ij}^{xy,RA} \le (1 - s_{ij}^{i,NO,SS})M + f_i^{xy,RA}, \forall ij \in \gamma_I^{SS}, ij \notin \gamma_I^B \tag{2}$$

$$-(1 - s_{ij}^{i,xy,CB})M + f_i^{xy,RA} \le f_{ij}^{xy,RA} \le (1 - s_{ij}^{i,xy,CB})M + f_i^{xy,RA}, \forall ij \notin \gamma_I^{SS} \tag{3}$$

$$f_{ij}^{xy,RA} = f_i^{xy,RA}, \forall ij \notin \gamma_I^S, ij \notin \gamma_I^B \tag{4}$$

$$\sum_{ij \in \gamma_I^B} s_{ij}^{i,NO,CB} + \sum_{ij \in \gamma_J^B} s_{ij}^{j,NO,CB} - 1 = \sum_{ij \in \gamma_I^B} s_{ij}^{i,xy,CB} + \sum_{ij \in \gamma_J^B} s_{ij}^{j,xy,CB} \tag{5}$$

$$f_i^{xy,RA} = 1, \forall i \in \Psi^{SUB} \tag{6}$$

$$1 - f_i^{xy,RA} - x_i^{UPS} \le p_i^{xy} \le 1 - x_i^{UPS}, \forall i \in \Psi^{LN} \tag{7}$$

Here, $f_i^{xy,RA}$ and $f_{ij}^{xy,RA}$ represent the virtual power flow variable in the *RA* stage. If the failure of line *xy* will cause a power outage of node *i*, $f_i^{xy,RA} = 0$; otherwise, $f_i^{xy,RA} = 1$. If the failure of line *xy* will cause a power outage of line *ij*, $f_{ij}^{xy,RA} = 0$; otherwise $f_{ij}^{xy,RA} = 1$. The logical variable $s_{ij}^{i,xy,CB}$ represents the state of the circuit breaker which is on or off at node *i* when line *xy* is in the fault state; $s_{ij}^{i,xy,SS}$ represents the state of the segmented switch which is on or off at node *i* when line *xy* is in the fault state; $s_{ij}^{i,NO,CB}$ represents the state of the circuit breaker which is on or off at node *i* when line *xy* is in the normal state; $s_{ij}^{i,NO,SS}$ represents that the segmented switch is on or off at node *i* when line *xy* is in the normal state; $p_i^{xy}$ is a logical variable that represents whether node *i* is powered off due to a line *xy* fault; $x_i^{UPS}$ is a logical variable that indicates whether node *i* is installing UPS.

In this model, (1) represents the source of the virtual power flow during the *RA* stage; (2)–(4) represent that after a line *xy* fault, the virtual power flow is transmitted through nodes and lines and isolated through circuit breaker tripping; (5) indicates that there is only one circuit breaker that will trip after a line *xy* fault; (6) indicates that the substation nodes will not be affected by line *xy* fault events; (7) represents a variable constraint. If node *i* is not installed with UPS and is affected by a fault, the value is 1; otherwise, it is 0.

### 3.2. Virtual Power Flow Model in SSI Stage

After the *RA* stage, the segmented switch preliminarily isolates the faulty line and restores the upstream load power supply from the main power supply. The virtual power flow model in the *SSI* stage is shown in (8)–(14).

$$f_{xy}^{xy,SSI} = 0 \tag{8}$$

$$-(1 - s_{ij}^{i,xy,FI})M + f_i^{xy,SSI} \le f_{ij}^{xy,SSI} \le (1 - s_{ij}^{i,xy,FI})M + f_i^{xy,SSI} \tag{9}$$

$$-(1 - s_{ij}^{i,xy,SS})M + f_i^{xy,SSI} \le f_{ij}^{xy,SSI} \le (1 - s_{ij}^{i,xy,SS})M + f_i^{xy,SSI}, \forall ij \in \gamma_I^{SS} \tag{10}$$

$$f_{ij}^{xy,SSI} = f_i^{xy,SSI}, \forall ij \notin \gamma_I^{SS} \tag{11}$$

$$f_i^{xy,SSI} = 1, \forall i \in \Psi^{SUB} \tag{12}$$

$$q_i^{xy,SSI} = f_i^{xy,SSI} - x_i^{UPS} \tag{13}$$

$$k_{ij}^{xy,SSI} \leq f_{ij}^{xy,SSI} \leq 1 \tag{14}$$

Here, $f_i^{xy,SSI}$ and $f_{ij}^{xy,SSI}$ represent the virtual power flow variable in the *SSI* stage, and the definition of the variable is similar to that in the *RA* stage. $q_i^{xy,SSI}$ represents whether the load point can be restored after the fault is isolated by the segmented switch during the *SSI* stage; $k_{ij}^{xy,SSI}$ represents the state variable of the line *ij* during the *SSI* stage, when the line is a path, $k_{ij}^{xy,SSI} = 1$; otherwise, $k_{ij}^{xy,SSI} = 0$; $s_{ij}^{i,xy,FI}$ represents the logical variable that when line *xy* is in the fault state, the fault indicator is in the on state or off state. *M* represents a constant with a large value.

In this model, (8) represents the virtual power flow source during the *SSI* stage; (9)–(11) represent the virtual power flow propagation process during the *SSI* stage; (12) represents that the substation nodes are not affected by the virtual power flow during the *SSI* phase; (13) represents the logical variable that determines whether load power at node *i* can be restored during the *SSI* stage through segmented switch action. If load power can be restored during the *SSI* stage, then $q_i^{xy,SSI} = 1$; otherwise, $q_i^{xy,SSI} = 0$; (14) represents the operating status of line *ij* during the *SSI* stage.

*3.3. Virtual Power Flow Model in LT Stage*

After the *SSI* stage, the fault indicator is combined with manual line inspection to locate the fault, further isolate the fault, and transfer the load with power outage to other power sources in conjunction with the SOP. The virtual power flow in the *LT* stage starts from the faulty components and propagates through the state variables of nodes and lines until the segmented switches pass through. The state variables of nodes and lines are used to represent the recoverable power supply range in the *LT* stage and the continuous power outage range in the *LT* stage.

The virtual power flow model in the *LT* stage is shown in (15)–(21):

$$f_{xy}^{xy,LT} = 0 \tag{15}$$

$$-(1 - s_{ij}^{i,xy,SS} - s_{ij}^{i,xy,SOP})M + f_i^{xy,LT} \leq f_{ij}^{xy,LT} \leq (1 - s_{ij}^{i,xy,SS} - s_{ij}^{i,xy,SOP})M + f_i^{xy,LT} \tag{16}$$

$$f_{ij}^{xy,LT} = f_i^{xy,LT}, \forall ij \notin \gamma_I^{SS} \tag{17}$$

$$f_i^{xy,LT} = 1, \forall i \in \Psi^{SUB} \tag{18}$$

$$q_i^{xy,LT} = f_i^{xy,LT} - f_i^{xy,SSI} \tag{19}$$

$$k_{ij}^{xy,LT} \leq f_{ij}^{xy,LT} \leq 1 \tag{20}$$

$$q_i^{xy} = q_i^{xy,SSI} + q_i^{xy,LT} + x_i^{UPS} \tag{21}$$

The definition of related variables is consistent with the previous text, and the specific meanings of superscripts, subscripts, and constraints can also refer to the previous text.

## 4. MILP-Based Active Distribution Network Reliability Evaluation Method

This section considers the states of circuit breakers, fault indicators, SOPs, segmented switches, UPS, and other equipment participating in the fault recovery process. Based on the improved virtual power flow model mentioned above, an ADN reliability evaluation method based on MILP is established. Due to the fact that the proposed reliability model belongs to MILP and the MILP problem has global convergence, global optimality can be guaranteed. Below is a detailed introduction to the MILP-based ADN reliability evaluation method.

### 4.1. Objective Function of ADN Reliability Evaluation

Firstly, the power outage time at the load node can be calculated according to Equation (22):

$$U_i = \sum_{m=1}^{N} \lambda_m \tau_m \tag{22}$$

Here, $U_i$ represents the shutdown time of load node $i$ due to component failure in that year; $\tau_m$ represents the component failure repair time under the $m$-th fault scenario; $\lambda_m$ represents the annual failure frequency of components under the $m$-th fault scenario; $N$ represents the total number of fault scenarios.

This paper chooses the minimum system average interruption duration index (*SAIDI*) as the objective function, which can be described as (23).

$$obj : \min SAIDI = \frac{\sum\limits_{i=1}^{n} \sum\limits_{m=1}^{N} U_i N_i}{\sum\limits_{i=1}^{n} N_i} \tag{23}$$

### 4.2. Other Constraints

(1)　Constraints of improved virtual power flow

The improved virtual power flow constraints are given in Equations (1)–(21) in Section 3. Its purpose is to linearly represent the state variables of nodes, lines, and equipment through virtual power flow, providing a model basis for the analytical expression of reliability indicators.

SOP mainly utilizes fully controlled power electronic devices for control. This section takes the back-to-back voltage source converter as an example and selects the PQ-$V_{dc}Q$ control mode. The back-to-back voltage source converter (VSC) of the SOP consists of a DC capacitor and two converters, i.e., VSC1 and VSC2, shown in Figure 1. Due to the DC isolation effect of the SOP, it can be considered a normally open breakpoint under normal operation.

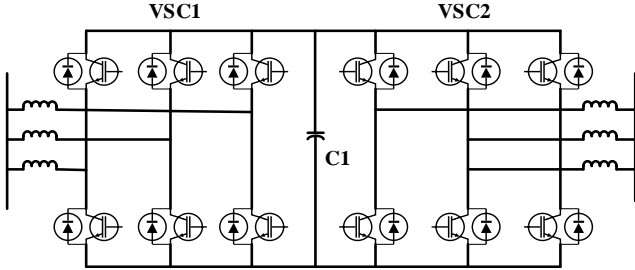

**Figure 1.** Circuit topology of SOP.

Under normal operation, one of the converters is in active reactive power (PQ) control mode; the other converter is in the $V_{dc}Q$ control mode, controlling the stability of the DC voltage. When the AC-side system of the converter malfunctions, the faulty-side converter is in the AC voltage frequency (Vf) control mode to achieve voltage support, while the other-side converter is in the $V_{dc}Q$ control mode to achieve uninterrupted power supply on the non-faulty side.

There is a certain power loss during the state control and switching process of the *SOP*, but for larger distribution networks, the power loss of the *SOP* is very small and is ignored here. The *SOP* operation constraints are as follows (24)–(28):

$$P_{SOP}^{i,xy,ij} + P_{SOP}^{j,xy,ij} = 0 \tag{24}$$

$$\sqrt{\left(P_{SOP}^{i,xy,ij}\right)^2 + \left(Q_{SOP}^{i,xy,ij}\right)^2} \leq S_i^{SOP} \tag{25}$$

$$Q_{i,\min}^{SOP} \leq Q_{SOP}^{i,xy,ij} \leq Q_{i,\max}^{SOP} \tag{26}$$

$$s_{ij}^{i,xy,SOP} - 1 \leq u_{ij}^{i,xy,SOP} + u_{ij}^{j,xy,SOP} \leq x_{ij}^{SOP} \tag{27}$$

$$-Mu_{ij}^{i,xy,SOP} \leq U_i - U_{SOP}^{ref} \leq Mu_{ij}^{i,xy,SOP} \tag{28}$$

Here, $P_{SOP}^{i,xy,ij}$ represents the active power output by the SOP at node $i$ when line $xy$ is in the fault state; $Q_{SOP}^{i,xy,ij}$ represents the reactive power output by the SOP at node $i$ when line $xy$ is in the fault state; $S_i^{SOP}$ represents the converter capacity of SOP at node $i$; $Q_{i,\max}^{SOP}$ and $Q_{i,\min}^{SOP}$ represent the maximum and minimum reactive power output of SOP, respectively; $U_{SOP}^{ref}$ represents the square reference value of the voltage at the fault side during the fault recovery process; $u_{ij}^{i,xy,SOP}$ represents whether the port converter of SOP on line $ij$ is in $V_{dc}Q$ control mode.

In this model, (24) represents the active power balance constraint of the converter; (25) represents the power constraints of the converters; (26) represents the output reactive power constraint of the converter; (27) indicates that only one converter at both ends of the SOP is in $V_{dc}Q$ control mode after a failure; and (28) indicates that the voltage at the fault side is the reference voltage value of SOP.

Using the linearization method, (25) can be transformed into linear constraints. The detailed content can be referred to in reference [24] and will not be elaborated here.

(2) Other operational constraints

$$-M(1 - q_i^{xy}) \leq P_i^{xy} + P_{DG}^{i,xy} + P_{SOP}^{i,xy,ij} - P_d^i(1 - x_i^{UPS}) \leq M(1 - q_i^{xy}) \tag{29}$$

$$-M(1 - q_i^{xy}) \leq Q_i^{xy} + Q_{DG}^{i,xy} + Q_{SOP}^{i,xy,ij} - Q_d^i(1 - x_i^{UPS}) \leq M(1 - q_i^{xy}) \tag{30}$$

$$P_i^{NO} = P_d^i - P_{DG}^{i,NO} \tag{31}$$

$$Q_i^{NO} = Q_d^i - Q_{DG}^{i,NO} \tag{32}$$

$$-M(1 - k_{ij}^{xy}) + 2(r_{ij}P_{ij}^{xy} + x_{ij}Q_{ij}^{xy}) \leq U_i^{xy} - U_j^{xy}$$
$$\leq M(1 - k_{ij}^{xy}) + 2(r_{ij}P_{ij}^{xy} + x_{ij}Q_{ij}^{xy}), \forall xy \in \gamma \cup xy = NO \tag{33}$$

$$U_i^{xy} = U_{\max} \quad , \forall i \in \Psi^{SUB} \tag{34}$$

$$U_{\min} \leq U_i^{xy} \leq U_{\max} \quad , \forall i \in \Psi^{LN} \tag{35}$$

$$P_{DG}^{i,xy}, Q_{DG}^{i,xy} \leq \alpha_i^{DG} S_{\max}^{DG}, \forall i \in \Psi^{LN} \tag{36}$$

$$P_{DG}^{i,xy} \pm Q_{DG}^{i,xy} \leq \sqrt{2}\alpha_i^{DG} S_{\max}^{DG}, \forall i \in \Psi^{LN} \tag{37}$$

$$k_{ij}^{xy} = s_{ij}^{i,xy,SOP} + s_{ij}^{j,xy,SOP} + s_{ij}^{i,xy,SS} + s_{ij}^{j,xy,SS} \tag{38}$$

$$-Mk_{ij}^{xy} \leq P_{ij}^{xy} \leq Mk_{ij}^{xy}, \forall xy \in \gamma \cup xy = NO \tag{39}$$

$$-Mk_{ij}^{xy} \leq Q_{ij}^{xy} \leq Mk_{ij}^{xy}, \forall xy \in \gamma \cup xy = NO \tag{40}$$

$$P_{ij}^{xy} \pm Q_{ij}^{xy} \leq \sqrt{2}S_{ij}^{C}, \forall xy \in \gamma \cup xy = NO \tag{41}$$

$$-S_{ij}^{C} \leq P_{ij}^{xy}, Q_{ij}^{xy} \leq S_{ij}^{C}, \forall xy \in \gamma \cup xy = NO \tag{42}$$

$$P_f^{xy} = P_{tr^f}^{xy}, \forall f \in \Psi^F, tr^f \in \Psi^{LN}, \forall xy \in \gamma \cup xy = NO \tag{43}$$

$$Q_f^{xy} = Q_{tr^f}^{xy}, \forall f \in \Psi^F, tr^f \in \Psi^{LN}, \forall xy \in \gamma \cup xy = NO \tag{44}$$

$$-\sqrt{2}S_f^{C} \leq P_f^{xy} \pm Q_f^{xy} \leq \sqrt{2}S_f^{C}, \forall xy \in \gamma \cup xy = NO \tag{45}$$

$$-S_f^{C} \leq P_f^{xy}, Q_f^{xy} \leq S_f^{C}, \forall xy \in \gamma \cup xy = NO \tag{46}$$

$$k_{ij}^{xy} \leq x_{ij}^{EL}, \forall xy \in \gamma \cup \forall xy = NO \tag{47}$$

Here, $P_i^{xy}$ and $Q_i^{xy}$ represent the active and reactive power injected by node $i$, respectively; $P_{ij}^{xy}$ and $Q_{ij}^{xy}$ represent the active and reactive power flowing through the line $ij$, respectively. $P_{DG}^{i,xy}$ and $Q_{DG}^{i,xy}$ represent the active and reactive power emitted by DG at node $i$, respectively; $P_d^i$ and $Q_d^i$ represent the active and reactive power of the load at node $i$, respectively; $U_i^{xy}$ represents the square of the voltage at node $i$; $U_{\min}$ and $U_{\max}$ represent the minimum and maximum squared value of node voltage, respectively; $S_{\max}^{DG}$ represents DG capacity; $a_i^{DG}$ represents whether node $i$ is connected to DG; $k_{ij}^{xy}$ represents the connected variable of the line $ij$ after the fault; if the line is connected, then $k_{ij}^{xy} = 1$; otherwise, $k_{ij}^{xy} = 0$; $S_{ij}^{C}$ represents the line capacity; $P_f^{xy}$ and $Q_f^{xy}$ represent the active and reactive power output by the transformer; $tr^f$ represents the outgoing line to which feeder $f$ belongs; $x_{ij}^{EL}$ is a logical variable that represents the existence of circuit $ij$ in the system.

In this model, (29) and (30) represent the expressions of active and reactive power injected by nodes in fault scenarios, respectively. These constraints use the big-M method to relax, and M represents a constant; (31) and (32) represent the injection of active and reactive power by nodes during normal system operation; (33) represents the node and line power flow constraints; (34) represents the voltage constraint of the substation node; (35) represents the voltage constraint at the load node; (36) and (37) represent the DG output

constraints; (38) represents the line connectivity constraints after network reconstruction; (39)–(42) represent the constraints on active and reactive power of the line; (43)–(46) represent the constraints on active and reactive power emitted by the transformer; (47) indicates that line *ij* can be connected only when line *ij* exists.

## 5. Case Study

In order to evaluate the effectiveness of the ADN reliability evaluation model proposed in this paper, an improved 33-node testing system is used for case analysis. This model is based on MATLAB 2019b programming and solved using CPLEX 12.9. All calculation results were analyzed on an Inter Core i5-7300 2.4 GHz processor and a computer with 16 GB of memory. The topology of the IEEE 33-node testing system is shown in Figure 2. The load level of the power system is time-varying. The electricity consumption in summer and winter is generally greater than that in spring and autumn. If a fixed load value is used, it will lead to inaccurate reliability evaluation results. On the other hand, the working state of the equipment is relatively stable, so its failure rate is considered a fixed value. The load level, fault data, and the number of customers used in this paper are selected from reference [25]. The failure rate of lines in the system is set to 0.1/(km/year). Uncertainty at the load side is represented by three different levels, namely, 100%, 83%, and 70% of the peak load of the node. The duration of each load level is 1000 h/year, 5760 h/year, and 2000 h/year, respectively. The division of specific time periods can be randomly generated using Monte Carlo simulation methods. On the other hand, in order to match the test data with the actual situation, high load level usually appears in the noon and afternoon of summer and winter, and low load level usually appears in the morning and late night of spring and autumn. In practical applications, power grid operators can determine the final load level based on the previous year's load level while considering the load growth rate. The SOP action time is set to 0.1 h, and the unit inspection time for overhead lines is set to 5 min/km. The SOP capacity is set to 1 MVA.

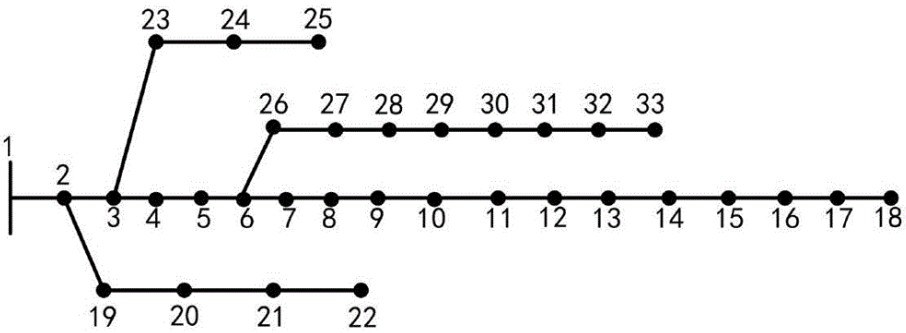

**Figure 2.** Topology of the IEEE 33-node testing system.

### 5.1. Reliability Evaluation Comparison under Different Configuration Cases

This section considers the impact of different device configurations on system reliability and uses an improved 33-node system for case analysis to verify the effectiveness of the proposed reliability evaluation method.

The following seven different cases are set to analyze reliability indicators under different configurations. In case 7, UPS is installed at nodes 14 and 18, a wind turbine is installed at nodes 11 and 25, and photovoltaic is installed at node 19.

Case 1: Without considering UPS and DG, the circuit breaker is configured at the substation node with a small number of segmented switches in the middle of the feeder line and no fault indicators. The tie line is equipped with interconnection switches;

Case 2: Without considering UPS and DG, the circuit breaker is configured at the substation node with a small number of segmented switches and fault indicators in the middle of the feeder line. The tie line is equipped with interconnection switches;



Case 3: Without considering UPS and DG, the circuit breaker is configured at the substation node. All branches of the feeder line are equipped with segmented switches, and a small number of fault indicators are installed in the middle of the feeder line. The tie line is equipped with interconnection switches;

Case 4: Without considering UPS and DG, the circuit breaker is configured at the substation node with a small number of segmented switches and fault indicators in the middle of the feeder, and the SOP is configured on the interconnection line;

Case 5: Without considering UPS and DG, the circuit breaker is configured at the substation node, and all branches of the feeder are equipped with segmented switches. A small number of fault indicators are installed in the middle of the feeder, and the SOP is configured on the interconnection line;

Case 6: Without considering UPS and DG, the circuit breaker is configured at the substation node and all branches of the feeder are equipped with segmented switches. A small number of fault indicators are configured in the middle of the feeder, and the SOP is configured on the interconnection line;

Case 7: Consider UPS and DG configurations; circuit breakers are configured at the beginning of all branches and segmented switches are configured for all feeder branches. A small number of fault indicators are configured in the middle of the feeder, and the SOP is configured for the interconnection line.

From Figure 3, it can be seen that compared to case 1, case 2 has added a fault indicator, which reduces SAIDI by 25.9% (0.7145 h/year) and EENS by 25.9% (32.44 MWh/year). The reason is that the fault location using fault indicator can reduce the patrol time, effectively reducing the outage time and range. As the number of segmented switches increases, compared to case 2, the SAIDI and EENS of case 3 significantly decrease by 64.92% (1.3274 h/year) and 65.43% (60.74 MWh/year), respectively. Compared to case 4, case 5 significantly reduces SAIDI and EENS by 66.49% (1.2925 h/year) and 67.57% (59.63 MWh/year), respectively. With the replacement of switches with a SOP, the SAIDI and EENS of case 4 decrease by 4.93% (0.1008 h/year) and 4.93% (4.58 MWh/year), respectively. Similarly, compared to case 3, case 5 reduces SAIDI and EENS by 9.19% (0.0659 h/year) and 10.8% ( m), respectively. Through analysis, it can be concluded that the more segmented switches in the distribution network, the better the system reliability level. Replacing traditional switches with SOPs can effectively reduce SAIDI and EENS. Compared with cases 2, 3, 4, and 5, case 6 has smaller SAIDI and EENS, indicating that as the number of circuit breakers installed increases, SAIDI and EENS both decrease; in other words, installing more circuit breakers can reduce the range of fault impact. Compared with case 6, case 7 has smaller SAIDI and EENS. It can be seen that the connection of UPS has a significant impact on reducing power outage time. The results indicate that the reliability evaluation method proposed in this paper is applicable to various equipment configuration modes, and reasonable equipment configuration has a beneficial effect on the reliability of distribution networks.

From Figure 4, it can be seen that the power outage frequency of load node in case 3 and case 5 is the same. The reason is that the power outage frequency is determined by the system structure during normal operation, and the switches and SOP are approximately equivalent to open circuits during normal operation. Therefore, using SOPs to transfer power will not affect the power outage frequency. It can also be seen that considering SOPs for load transfer can further improve system reliability. For example, load nodes 12, 21–25, etc., will not be able to restore power supply. The reason is that compared to contact switches, a SOP can provide power support and has a shorter transfer time, thereby reducing power outage time.

Compared to case 3 and case 5, the node outage frequency and outage time of case 7 have been significantly increased. It should be noted that due to the installation of UPS in nodes 16 and 20, the outage frequency and outage time are 0. The results indicate that considering the addition of circuit breakers, segmented switches, and UPS will significantly improve node reliability, verifying the effectiveness of the proposed method.

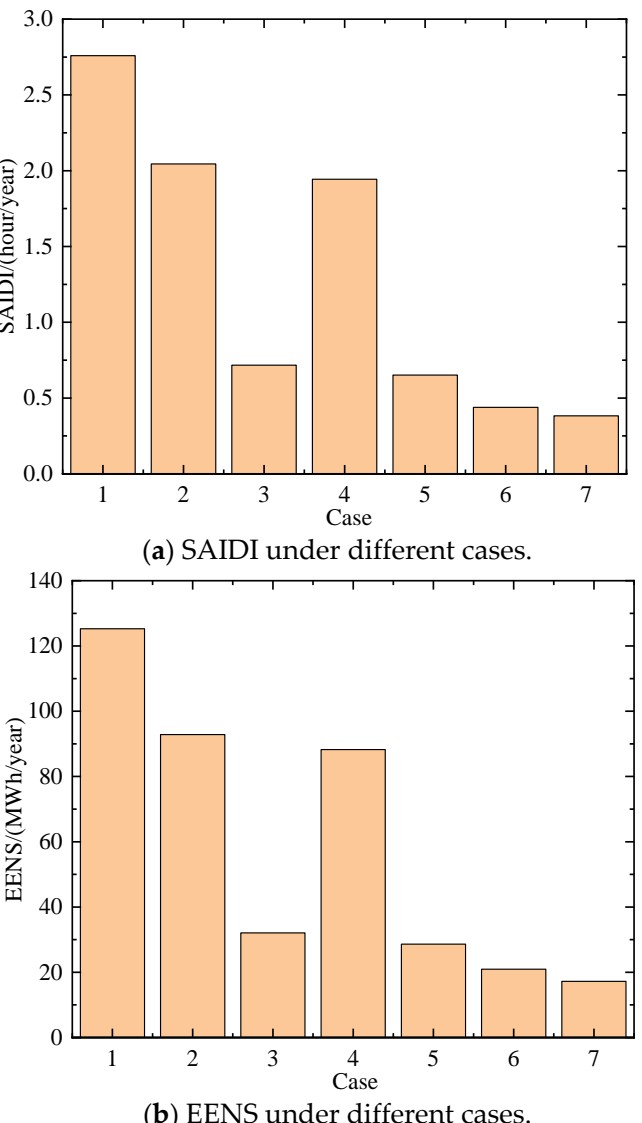

(**a**) SAIDI under different cases.

(**b**) EENS under different cases.

**Figure 3.** Results under different cases.

*5.2. Comparative Analysis of Existing Methods*

In order to further test the effectiveness of the proposed reliability evaluation model, the proposed method is compared with the method suggested in reference [16]. The calculation results of system indicators obtained applying different methods are shown in Figure 5a,b.

From Figure 5, it can be seen that in the modified IEEE 33-node system, the reliability results obtained by the proposed method are consistent with those obtained by [16], verifying the correctness and scalability of this method. The value of SAIDI obtained by the proposed method and [16] are 0.3610 h/year and 0.4854 h/year, respectively. The value of the EENS indicator obtained by the proposed method and [16] are 16.57 MWh/year and 23.43 MWh/year. Due to the comprehensive consideration of the effects of circuit breakers, segmented switches, fault indicators, UPS, and SOPs, compared with the results obtained by the method in reference [16], in the modified 33-node testing system, the SAIDI calculated by the proposed model decreases by 25.6%, and the EENS decreases by 29.3%. Therefore, considering the impact of various equipment comprehensively has a significant beneficial effect on reliability improvement.

On the other hand, under the same grid structure, the reliability indicators obtained by the method are the same as those obtained by the simulation method. In the mod-

ified IEEE 33-node testing system, compared to the simulation method, the proposed method reduces the solving time by 36.4%, proving that the proposed method has high computational efficiency.

To further show the applicability of the proposed method, a large-scale testing system is given, i.e., the IEEE 123-node testing system. The topology of the IEEE 123-node testing system is shown in Figure 6. Similarly, seven different cases are set, and the detailed case information is consistent with the modified IEEE 33-node testing system mentioned earlier. The detailed calculation results are shown in Table 1. It can be clearly seen from Table 1 that the proposed method (i.e., case 7) can effectively reduce power outage frequency and EENS value, significantly improving the operational reliability of the system.

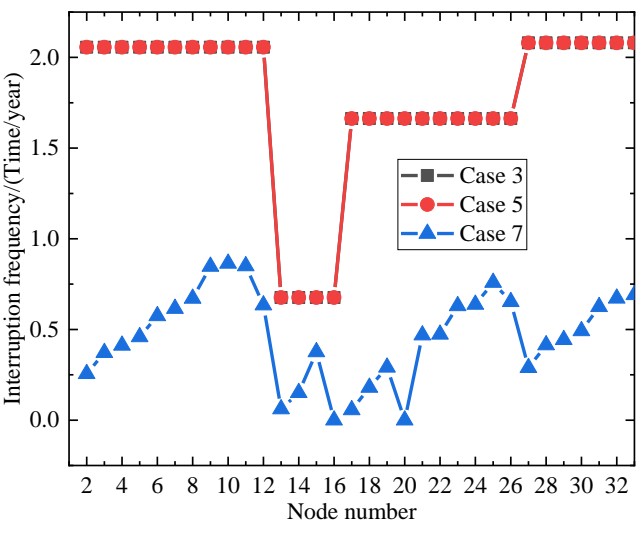

(**a**) Interruption frequency under different cases.

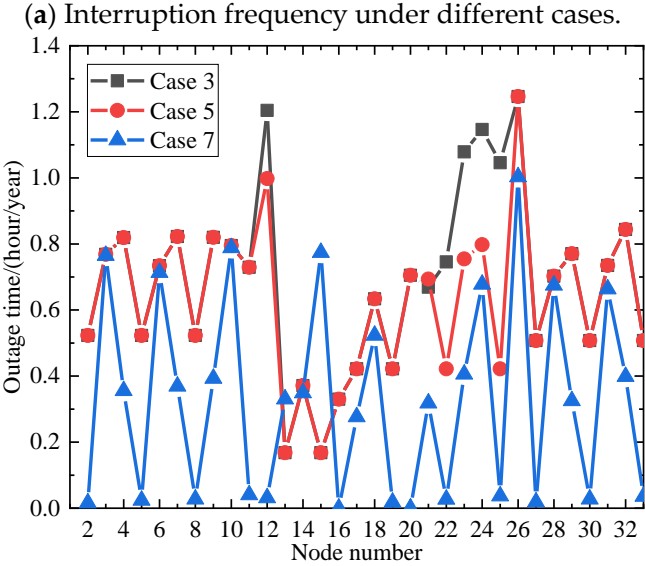

(**b**) Outage time under different cases.

**Figure 4.** Detailed information under different cases.

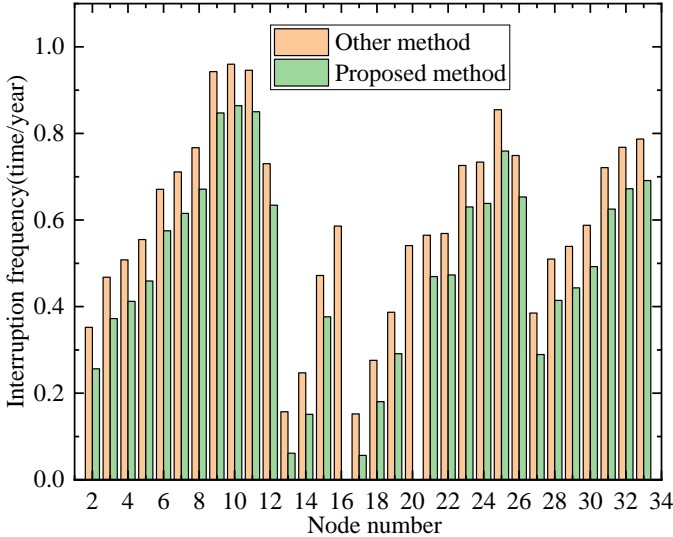

(**a**) Interruption frequency comparison with other method.

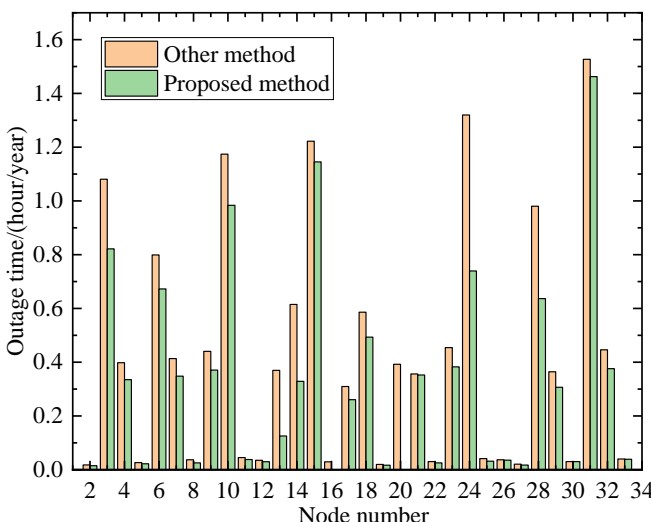

(**b**) Outage time comparison with other method.

**Figure 5.** Results comparison with other method [16].

**Table 1.** Reliability indices of IEEE 123-node testing system for different cases.

| Case | SAIFI (Power Outage Frequency/Year) | EENS (MWh/Year) |
|---|---|---|
| Case 1 | 3.61 | 178.51 |
| Case 2 | 3.43 | 135.07 |
| Case 3 | 3.17 | 50.53 |
| Case 4 | 3.65 | 117.81 |
| Case 5 | 3.43 | 37.78 |
| Case 6 | 0.94 | 33.01 |
| Case 7 | 0.82 | 28.40 |

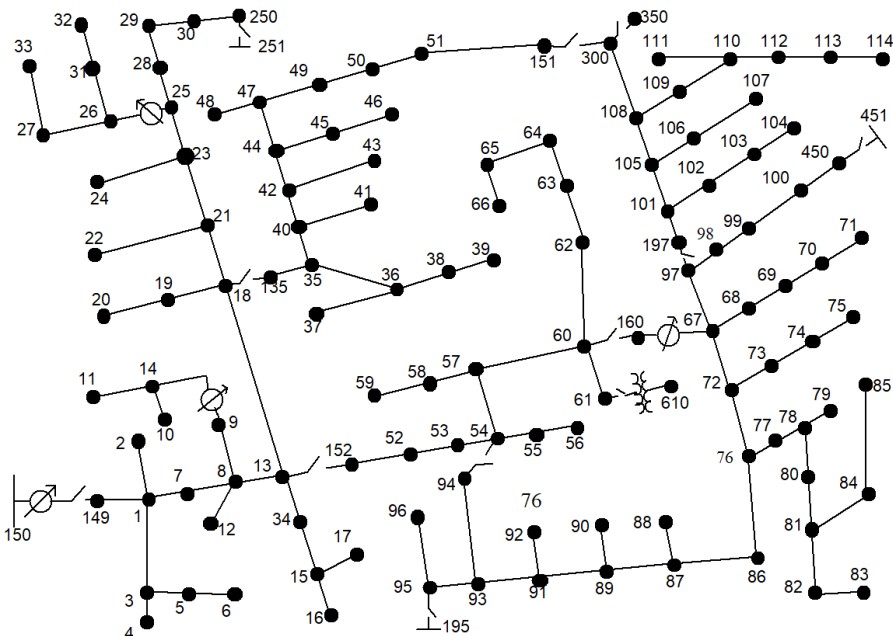

**Figure 6.** The topology of the IEEE 123-node testing system.

## 6. Conclusions

This paper proposes an ADN reliability evaluation method based on MILP which comprehensively considers circuit breakers, segmented switches, SOPs, fault indicators, and UPS. Firstly, by analyzing the impact of these different pieces of equipment on the load recovery process, an improved virtual power flow model is designed to describe the propagation process of fault effects. Then, a linear relationship between virtual power flow and reliability indicators is constructed; finally, a modified IEEE 33-node testing system is used to verify the feasibility of the proposed method.

(1) Compared with other methods that do not comprehensively consider the configuration of circuit breakers, segmented switches, SOPs, fault indicators, UPS, and other equipment, the reliability evaluation method proposed in this paper showed a decrease of 25.6% in SAIDI and 29.3% in EENS in the modified IEEE 33-node testing system. Due to the consideration of the combined mechanisms of multiple devices, the reliability results obtained by the proposed method are more in line with reality.

(2) The proposed reliability evaluation method has the same reliability evaluation results and higher solving efficiency compared to the simulation method. In the testing system, the proposed method reduces the solving time by 36.4%.

**Author Contributions:** Conceptualization, J.Z.; methodology, B.W.; software, H.M.; validation, Y.H.; formal analysis, Y.W.; investigation, J.Z.; resources, B.W.; writing—original draft preparation, Z.X. All authors have read and agreed to the published version of the manuscript.

**Funding:** This paper is supported by Science and Technology Project of State Grid Corporation Limited (Contract No. 5400-202119145A-0-0-00).

**Data Availability Statement:** Data are contained within the article.

**Conflicts of Interest:** Author Y.H. was employed by the company Electric Science Research Institute of State Grid Zhejiang Province Co., Ltd. The remaining authors declare that the research was conducted in the absence of any commercial or financial relationships that could be construed as a potential conflict of interest. The authors declare that this study received funding from Science and Technology Project of State Grid Corporation Limited (No. 5400-202119145A-0-0-00). The funder was not involved in the study design, collection, analysis, interpretation of data, the writing of this article or the decision to submit it for publication.

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
