# Peer review of "Reliability Evaluation of Cabled Active Distribution Network Considering Multiple Devices—A Generalized MILP Model"

_processes, doi:10.3390/pr11123404_

Round 1
Reviewer 1 Report
Comments and Suggestions for Authors
Active distribution networks (ADN) play a key role in modern electric power industry. The authors present a generalized MILP model for the reliability evaluation of these networks. The paper is interesting and deserves further discussion. The following suggestions must be considered by the authors:
1. The introduction section is well written; nonetheless, it can be complemented. For those references that apply, try to provide more details. Briefly mention their features (advantages and drawbacks) to have a clearer idea of the authors contribution.
2. One of the advantages of having a MILP model is the achievement of global optimal solutions. Although it is mentioned in the document, it must be emphasized in the introduction section.
3. Lines 294 to 296 are weird; it seems they belong to the MDPI template and not to the paper as such. Please check and correct.
4. To show the applicability of the proposed approach the authors are invited to test they approach in a bigger (or real) distribution test system.
5. In line 390 it must be “the proposed method is compared” using “is” instead of “are”.
6. The authors mention a comparison with the simulation method reported in [19] but the results are not presented in figure 5. It seems that only a comparting with [16] was carried out.
7. In figure 5 it is better to say “Proposed Method” than “The proposed method”
8. Numbering the conclusions is not necessary.
9. References do not follow the MDPI format.
Comments on the Quality of English LanguageEnglish is ok, only minor editing is required.
Reviewer 2 Report
Comments and Suggestions for Authors
1. The terminologies used in the equations are not all explained.
2. In lines 303-305, load levels, failure data are considered. What is the source for that? How is it standard?
3. How is the variation of loads changed?
4. For calculation of SAIDI, EENS and all, the number of customers are needed. Where is the data for that?
5. In section 5.2, for comparitive analysis, the values obtained by existing methods are not shown.
Comments on the Quality of English Language
The language seems fine.
Reviewer 3 Report
Comments and Suggestions for Authors
Paper presents an interesting study, but some issues such as follows, should be addressed:
- a reminder of the paper could be introduced in Section 1;
- in lines 133-135, please revise "The above processes all need to meet the power flow constraints after equipment operation, and the operating status of the equipment determines the power supply status of the load point."; meaning is unclear;
- in lines 149-151 it is stated " the reliability indicators of the load point can be directly calculated based on the given network topology, reliability parameters, and other data"; please specify which are the reliability indicators and reliability parameters;
-title of 4.1 section should be revised;
- please complete "The back-to-back voltage source converter" with (VSC1 and VSC2) in line 241 for a better readability;
- in lines 294-296 please revise or remove "Authors should discuss the results and how they can be interpreted from the perspective of previous studies and of the working hypotheses. The findings and their implications should be discussed in the broadest context possible. Future research directions may also be highlighted."
- in line 302, please revise: "The topology of the IEEE-33 node testing system is shown in Figure 3.7"; there is no such figure (is it Fig. 2?);
Comments on the Quality of English LanguageMinor misspelling or mistyping errors should be revised.
Round 2
Reviewer 1 Report
Comments and Suggestions for Authors
I have no further comments
Reviewer 2 Report
Comments and Suggestions for Authors
No further queries